# Evaluation of a Pilot Wellness Elective for Master of Public Health Students during the COVID-19 Pandemic

**DOI:** 10.3390/ijerph21050590

**Published:** 2024-05-03

**Authors:** Blaise Y. O’Malley, Edgard Etoundi-Ngono, Jianjun Hua, Joseph P. Nano, Catherine F. Pipas

**Affiliations:** 1The Dartmouth Institute for Health Policy & Clinical Practice, Dartmouth Geisel School of Medicine, Hanover, NH 03755, USA; blaise.o’malley@students.jefferson.edu (B.Y.O.); 2Sidney Kimmel Medical College, Thomas Jefferson University, Philadelphia, PA 19107, USA; 3Department of Dartmouth Information, Technology and Consulting, Dartmouth Geisel School of Medicine, Hanover, NH 03755, USA

**Keywords:** student wellness, graduate student wellbeing, COVID-19 pandemic, public health students, university wellness curriculum

## Abstract

Background: Graduate student wellbeing is a public health issue in the United States. The COVID-19 outbreak exacerbated the mental health burden on graduate students worldwide. Culture of Wellness (PH 104) is a 2-week wellbeing elective course that teaches evidence-based wellbeing strategies for graduate students at a university in the United States. Our study aimed to evaluate the impact of this pilot wellbeing elective on Master of Public Health students’ mental health and wellness during the COVID-19 pandemic. Methods: Participants included 22 Master of Public Health students from the class of 2021 at a university in the United States. We provided a pre-course survey to students that assessed their perception of their own personal wellbeing, their knowledge about various wellbeing strategies, and their confidence in applying 13 wellbeing strategies before taking the course. Post-course students completed the same survey following course completion, as well as a matching evaluation and a five-month follow up survey. Results: Of the 13 strategies taught, students reported significant improvements in their ability to apply 10 strategies. There was a significant increase in self-reported emotional and physical wellbeing, as well as a significant decrease in burnout. Five months post-course, more than three quarters of respondents used strategies taught in the course on a weekly basis or more. Limitations: This pilot study is limited by its small sample size, which may restrict the generalizability of the findings. Conclusions: The PH 104 Culture of Wellness course was effective in improving graduate students’ wellbeing and confidence in applying wellbeing strategies

## 1. Introduction

Graduate student wellbeing, including mental and social health, has been a major problem in the United States and around the world for several years [1,2,3]. Graduate students face numerous challenges and stressors including academic pressures, job uncertainty, fear of failure, and negative social comparison [4]. Research has shown that graduate students are more likely to experience depression and anxiety compared with the general population and are at greater risk of suicidal thoughts [4]. In fact, a cross-sectional study by Evans et al. found that graduate students were over six times as likely to experience anxiety and depression as compared to the general population [1]. This study emphasized that many graduate students’ struggle to maintain a healthy work–life balance. Furthermore, Stallman et al. showed that many university students rely on unhealthy coping strategies during times of stress, including eating and negative self-talk [5].

The COVID-19 pandemic exacerbated mental health problems in both the general and student populations [6,7,8]. The cancellation of in-person classes, the transition to online learning, and social distancing measures reduced students’ social activity and contributed to feelings of loneliness and depression [7,9]. Additionally, university students reported difficulty concentrating as well as disrupted sleep patterns throughout the COVID-19 pandemic [9], both of which have been shown to contribute to poor academic performance [10,11]. Weak academic performance can worsen existing stress levels, negatively impacting student wellbeing. Given that graduate student wellbeing was a concern prior to the COVID-19 pandemic, it is now all the more urgent to address.

Several colleges and universities in the United States and beyond have created wellbeing initiatives (also referred to as wellness initiatives) in response to growing concerns regarding student mental health, both pre- and post-pandemic. These initiatives vary widely in scope and activities offered, with some initiatives being small-scale non-curricular events such as therapy dogs [12] and others being larger undertakings, such as wellness-focused courses embedded into the curriculum [13]. Many schools have piloted wellness programs or courses to evaluate their impact on students and determine whether such courses can help to address student mental health problems [14]. For example, in a study by Young et al. at Melbourne University in Australia, researchers assessed the impact of incorporating a wellbeing program in undergraduate psychology classes [14]. This program included two to three in-class or take-home interventions per week over a 6-week period. In-class activities included mindfulness meditation and self-compassion letter writing, while take-home activities included exercises on acting in line with values, as well as daily acts of kindness. The program measured their success based on the 14-item Mental Health Continuum Short Form scale, which showed that wellbeing strategies acted as a buffer against academic stress [14]. In another study, researchers at the Albert Einstein College of Medicine in the United States conducted a pilot of LAVENDER, a positive psychology program designed to help students maintain their wellbeing through self-compassion and emotional awareness, among other skills [13]. The intervention was delivered to 157 third-year medical students, with 76% of them ultimately agreeing that the skills taught in the intervention were useful.

Most university wellbeing programs have largely been focused on undergraduate and medical students [13,15,16]. Furthermore, best practice has not been established, with large variation in the content and target outcomes across these courses. As such, there is a gap in the literature on programs that focus specifically on graduate students as well as programs with comprehensive coverage of wellness skills and outcomes.

Although further research is needed on wellbeing programs for graduate students of all disciplines, the need to address graduate student mental health is especially pressing for students pursuing a degree in public health. Many public health graduate students pursue careers in medicine, nursing, hospital administration, or epidemiology, all of which are a part of the healthcare system. As such, major stressors have always been associated with and expected in the field of public health [17]. However, the COVID-19 pandemic exacerbated this association substantially, contributing to significant burnout amongst healthcare workers [18]. Burnout has been associated with poor mental and physical health outcomes, as well as poor patient safety outcomes [18,19]. Wellbeing courses that teach coping strategies, stress management, improvement processes, leadership, and wellness skills equip students with tools that could prevent burnout and potential mental health issues commonly experienced by healthcare and public health professionals before they become professionals themselves [20,21].

Recognizing the need to address this issue, a public health school in the United States offered an elective wellbeing course to its Master of Public Health (MPH) students belonging to the class of 2021 in December 2020. This two-week course, Culture of Wellness (PH 104), aimed to advance students’ capacity to model evidence-based wellbeing strategies and to collaborate with colleagues in creating cultures of wellbeing. The purpose of our study was to evaluate the impact of this wellbeing elective on Master of Public Health students’ mental health and wellbeing in the midst of the COVID-19 pandemic.

## 2. Materials

### 2.1. Course Description

In response to the COVID-19 pandemic, the “Culture of Wellness” (PH 104) course was offered to students from the class of 2021 as an online elective course during the 2020–2021 “Winterim” session (a two-week period between the fall and winter terms). Students were offered seven elective courses and were required to choose at least one. Out of 67 students in the MPH residential program, 22 enrolled in and completed this course.

Participants met via Zoom three times a week for two-hour sessions, for a total of 12 h of classroom time sessions between 30 November 2020 and 11 December 2020. The course learning objectives were aligned with assessment tools (Table 1). The course content was based on the 12 chapters in the book *A Doctor’s Dozen: 12 Strategies for Personal Health and A Culture of Wellness* [22], with each chapter telling the story of a different case and each case focusing on a different wellness-related strategy. This book provides 12 strategies for self-care: mindfulness, self-reflection, resilience, narrative writing, healthy eating, exercise, relationships, time management, leading change/emotional intelligence, prioritizing purpose, cognitive reframing, and appreciative inquiry.

Each class session focused on two cases, and therefore two strategies, from the book.

Students who completed the course received 0.5 credits in the MPH program for a total of 50 h of instruction, including 12 h in class, 12 h of reading and reflection, 12 h on the personal health improvement plan (PHIP), and 14 h on the Culture of Wellness Team project. The format of each session consisted of a brief welcome and mindfulness exercise, a check-in on the PHIP and team projects, and a discussion of each of two cases highlighting the application of two wellbeing strategies, followed by breakout room small group discussions, and finally closing remarks with the next steps for the following session. Each student was assessed by the course instructor based on participation and professionalism, pre-class readings and reflections, the PHIP project, and the team project (Appendix A).

During class discussions, the instructor highlighted evidence-based wellbeing strategies that were related to the readings, as well as methods that the students could use to incorporate each strategy into their personal lives and the school community.

### 2.2. Study Design and Data Collection

A presurvey on wellbeing was distributed to all participants prior to commencement of the course. Wellbeing outcomes were self-assessed based the following 4 validated wellbeing measures: burnout, quality of life, mindfulness, and perceived stress [23,24,25,26].

### 2.3. Pre-Post Wellbeing Strategies and Outcomes Surveys

An anonymous 13-question survey, set up through Qualtrics, was sent to all 22 participants before and after the Culture of Wellness course to determine their confidence in applying specific wellbeing strategies and their perception of their own personal wellbeing. Students were tracked using four survey methods. The four surveys offered throughout the course can be divided into four sections: (1) wellness strategies, (2) wellbeing outcomes, (3) course evaluations, and (4) a 5-month post-course follow-up.

Our pre- and post-surveys consisted of two parts: skill assessment and wellbeing assessment (Appendix A). For skill assessment, participants used one of the following scales: “Not at all”, “Minimally”, “Somewhat”, “Mostly”, and “Completely”. For wellbeing assessment, participants used various scales (Appendix A).

### 2.4. Baseline Student Wellness, Pre- and Post-COVID

To understand the impact of COVID-19 on students’ baseline wellbeing, we compared our Culture of Wellness (PH104) pre-course survey results with the identical pre-course survey from a public health course called Social Behavioral Determinants of Health (SBDoH) that was offered in Summer 2019 to students from the class of 2020 before the COVID-19 pandemic. This was a required 10-week course to all Master of Public Health students in the class of 2020. In contrast to the Culture of Wellness (PH104) course that was offered via Zoom to students from the class of 2021 in December 2020, SBDoH was offered in-person during the summer term and its course content included some of the personal wellbeing content that comprised the Culture of Wellness PH104 course.

The SBDoH course featured three main themes: personal wellbeing, public health and population health. The SBDoH course was completed by 54 first-year Master of Public Health students (all students in the class of 2020) compared to 22 students from the class of 2021 in the Culture of Wellness PH104 course. In response to COVID-19 in 2020, the “Culture of Wellness” elective was prioritized to focus solely on personal wellbeing and the aforementioned material. The wellbeing surveys, which were identical for the 2019 and 2020 courses, were given to all participants prior to and after each of the courses.

### 2.5. Course Evaluations and Projects

All students were sent an anonymous school-administered course evaluation after the course’s completion. Topics for both individual and team projects were categorized and analyzed to determine the students’ wellness priorities.

### 2.6. Five-Month Post-Course Follow Up

Participants received a follow-up survey 5 months after completing the course to assess their views on the benefits of the course after completing it. A total of 14 participants replied to the survey, which included 3 multiple choice questions and 1 open-ended question on the most valuable takeaway from the course (Table 2).

### 2.7. Analysis

Due to the data being sent via an anonymous link rather than a contact list that tracked each individual, we were only able to pair 10 pre- and post-survey results out of the possible 20 pairs; we conducted a paired *t*-test with these 10 pairs. The authors reviewed free text questions from the post-course evaluation and 5-month follow up survey. Lastly, we carried out independent *t*-test analyses comparing the 2019 and 2020 pre-course data. While for such a small sample size, a larger significance level might be preferred, based on the actual results from Section 3.2, we decided to set the significance level to 0.05 or 5% as the “boundary” between “significant” vs. “not significant”.

## 3. Results

### 3.1. Participants

A total of 22 students from the class of 2021 completed the Culture of Wellness (PH104) course in December 2020. A total of 20 students completed the pre-course wellbeing survey, all 22 students completed the post-course survey, 11 students completed the university standardized course evaluation, and 14 students completed the 5-month follow up survey. All participants are first-year graduate students on the Master of Public Health course.

### 3.2. Pre-Post Course Strategies and Wellbeing Surveys

All measures, with the exception of exercise (wellbeing outcomes and confidence in strategies), experienced a change in the desired direction. There was a statistically significant increase in the students’ confidence in their ability to apply the following wellbeing-enhancing strategies: mindfulness (*p* = 0.009), self-reflection (*p* = 0.002), resilience (*p* = 0.002), narrative writing (*p* = 0.004), social relationships (*p* = 0.047), time management (*p* = 0.005), leading change (*p* = 0.0001), prioritizing purpose (p = 0.01), cognitive reframing (*p* = 0.002), and appreciative inquiry (Table 3). Confidence in narrative writing showed the largest change, with a 54.5% increase from pre- to post-course (*p* = 0.004) (Table 3). Confidence in abilities to apply exercise strategies (*p* = 0.4), healthy eating (*p* = 0.3) and emotional intelligence (*p* = 0.2) did not show statistically significant changes.

In terms of quality-of-life wellbeing outcomes, there was a significant increase in self-reported mental wellbeing (14.5% change, *p* = 0.04), physical wellbeing (17.9% change, *p* = 0.006), and emotional wellbeing (28.3% change, *p* = 0.007). Of all quality-of-life outcomes, emotional wellbeing experienced the greatest change from pre- to post-course, increasing by 28.3% (Table 4). For mindfulness wellbeing outcomes, there was a significant increase in self-reported “feeling present in the moment” (19.4% change, *p* = 0.003) (Table 4). For burnout outcomes, there was a significant decrease in burnout experienced in the past month (−19.4% change, *p* = 0.01) (Table 4). For perceived stress outcomes, there was a significant decrease in the feeling that “there were difficulties piling up” so high they could not overcome them (−25.9% change, *p* = 0.005).

A few quality of life and perceived stress wellbeing outcomes displayed improvement but did not show statistical significance, such as self-reported “level of social activity” (14.8% change, *p* = 0.2), “spiritual wellbeing” (12.3% change, *p* = 0.1), “quality of life overall” (5.26% change, *p* = 0.2), “unable to control important things in their life” (−12.1%, *p* = 0.08), confidence in “abilities to handle personal problems” (0% change, *p* = 0.5), and the feeling that things were going their way (9.1% change, *p* = 0.1).

### 3.3. Course Evaluations and Student Projects

#### 3.3.1. Course Evaluations

Of the 22 students enrolled in the Culture of Wellness PH104 course, 11 completed the university standardized course evaluation. These anonymous evaluations were in an online survey format and were given to students following the course’s completion. The questions asked in the evaluation were standard across courses in the school’s MPH program.

Overall, all respondents rated the quality of the course as “very good” or “excellent”. Students listed the readings (55%, n = 6) and the group discussions and interactiveness during class (36%, n = 4) as the most effective components of the course.

When asked what aspects of the course were most effective, students responded as follows: “Referring to us as a team. I really liked being reminded that we are all in this together”; “Explaining that the time we were spending on our goals was built into the class helped me to ensure I was taking that time without feeling the pressure to be doing something else. Starting each class with a form of meditation was awesome.”; and “The content was relevant and timely (I hope everyone is allowed dedicated time to focus/learn about their wellbeing). And the organization of the course was excellent.”.

When asked how the course could be improved or enhanced, the majority of students (56%, n = 6) who responded to this question mentioned that the course should be longer, with one student stating that they felt it should be mandatory: “*I really liked every part of this seminar. Maybe have it be longer!*”; “*Sometimes it felt a little bit rushed—I just wish we had more time.*”; “*I would have liked more time to explore each topic. I know this was only a 2 week seminar but this could be turned into an elective so there would be more time to spend on each topic.*”…“*This was my favorite [graduate school] course—I was only sad it could not have been longer.*”; and “*Make it a longer course and make it mandatory for all of us.*”.

Lastly, the students’ response to the instructor was overwhelmingly positive, with 100% of respondents rating the effectiveness of the instructor’s teaching as “excellent”. When asked to give specific feedback to the instructor, students commented, “*Very understanding. Fostered an extremely positive environment. Welcomed us all by name each class. Called on everyone so that we all got a chance to participate.*”; “*[The instructor] was extremely encouraging and calm. She is a great role model for practicing wellness and was honest about her struggles with implementing it over the years.*”; and “*[The instructor] was organized, cared about her students, and carried out this course in an impeccable manner. I left each class feeling happier, hopeful, and grateful for all the things in my life.*”.

#### 3.3.2. Student Projects: Personal Health Improvement Plan and Team Project

All 22 students enrolled in the course created and implemented their own Personal Health Improvement Plan (PHIP) and participated in designing a team project. There were 6 teams in total, grouped by the nature or overarching theme of their PHIP. For example, all of the students who were focused on improving their sleep hygiene were placed in the same group and charged with identifying an additional collaborative team project.

Students were encouraged to choose a PHIP topic that was currently a health priority for them (Appendix A). Overall, the topics chosen encompassed all aspects of wellbeing, including social, spiritual, environmental, physical, emotional, occupational and intellectual. The most popular domain was physical wellbeing, with the most popular topic being exercise.

Team projects identified similar topics of health priorities but focused on community wellbeing rather than individual wellbeing. Both the PHIP and the community-based wellbeing goals in each project were framed as SMART goals (Specific, Measurable, Attainable, Relevant, and Time-based) (Appendix A).

### 3.4. Baseline Comparison of 2019 and 2020 Pre-Course Survey Wellbeing Outcomes

#### 3.4.1. Burnout

Burnout ratings increased significantly in 2020 compared to 2019. In the 2019 pre-course survey, 52% of participants reported feeling burnout, compared to 74% in 2020. This 22% change represents a significant increase (*p* = 0.04) in burnout ratings on the pre-course survey results from 2019 to 2020.

#### 3.4.2. Quality of Life

Using a 10-point scale with higher scores indicating better quality of life, the average quality of life rating was worse for the 2020 cohort, dropping from 8.83 in 2019 to 7.74 in 2020 (Table 5). Physical wellbeing was also lower in the 2020 cohort, with a mean of 7.53 compared to the 2019 mean of 7.71 (Table 5).

#### 3.4.3. Mindfulness

At baseline, 17% of 2019 participants reported feeling “not at all” present or in the moment during the past month compared with 0% of 2020 participants (Table 6). Of the participants in the 2020 course, 79% of the 2020 cohort reported feeling minimally or somewhat present in the moment during the past month. Additionally, 4% of students in 2019 felt completely present in the moment, compared with 0% in 2020.

#### 3.4.4. Perceived Stress

Responses to the Perceived Stress Scale questions ranged from 1 (never) to 5 (very often). The 2020 cohort reported feeling more like they were unable to control important things in their life (3.05 in 2020 vs. 3.01 in 2019), but also more confident about their ability to handle personal problems (Appendix A). Furthermore, the 2020 cohort reported that things went their way more frequently than the 2019 cohort did. The 2019 cohort also had higher (i.e., worse) scores when asked about difficulties piling up (Appendix A). However, none of the differences between the 2019 and 2020 results from this section of the survey were statistically significant.

### 3.5. Five-Month Post-Course Follow Up

A total of 14 out of the 22 course participants (64%) completed the post-course follow up survey. The results revealed that at least 71.5% of the student respondents considered wellbeing strategies to be moderately, very, or extremely important (Appendix A). The majority of respondents (78.5%) continued to apply the strategies learned in the course either weekly, daily, or more than once a day to improve their wellbeing 5 months post-course (Appendix A). In addition, 64.3% of the students felt either very confident or extremely confident in their ability to continue to apply the wellbeing strategies learned throughout the course (Appendix A).

#### Key Quotes from 5-Month Post-Course Survey

When asked what their most valuable takeaway from the course was, the students’ responses included the following: “Taking time for yourself”; “The value of mindfulness and expressing gratitude on a daily basis”; “In order to best serve others, you must serve yourself first, always”; “That we’re all struggling. It’s not just me or a few. It’s all of us”; “Taking a step back is important”; and “Failure is a part of the human experience and we have to embrace it if we want to embrace being fully human.”.

## 4. Discussion

### 4.1. Summary of Results

Our study found a significant increase in students’ confidence in their ability to apply 10 strategies after taking the course. In terms of wellbeing outcomes, we found a significant increase in self-reported well-being outcomes related to quality of life and mindfulness, and a significant decrease in self-reported burnout and perceived stress. Burnout ratings were significantly higher in the 2020 cohort compared to the 2019 cohort. Specific areas of quality of life that worsened from 2019 to 2020 included overall quality of life, spiritual wellbeing, and physical wellbeing. While mindfulness ratings worsened, more participants reported feeling “not at all present” in 2019 compared to 2020. In terms of perceived stress ratings, participants reported an overall higher level of confidence in their ability to handle stress in 2020 after taking Culture of Wellness (PH104) course. Additionally, the 5-month post-course follow up survey showed sustainable gains in the strategies taught and long-term retention of learnings.

### 4.2. Results in Context

Every year, graduate schools prepare Master of Public Health students to enter the public health field with specific knowledge and capabilities. Given the increasing mental health concerns and challenges experienced by workers in this field [18], it is clear that such knowledge and capabilities must include self-care strategies, as well as skills to improve an organization’s culture of wellbeing. Wellbeing strategies need to be taught to public health students before they enter the workforce, and the university setting provides the ideal place for this teaching.

This paper assessed the effectiveness of a two-week online wellbeing course focused on wellness strategies and wellbeing outcomes. The 2020 course, offered to public health graduate students at a university in the United States, engaged students in personal reflections, application of evidence-based wellbeing strategies through an individual Personal Health Improvement Project, and a community-based team project in which students held each other accountable for working towards specific wellness goals and advancing a culture of wellbeing. The 2020 pre- and post-course surveys, projects, and course evaluations demonstrated significant improvements in the majority of students’ wellbeing skills and outcomes, as well as high levels of satisfaction with the course. Additionally, the 5-month post-course follow up survey showed sustainable gains in the strategies taught and the long-term retention of learnings. This curriculum has been shown to have significant positive effects on public health students, equipping them with skills and strategies they can use as they enter an increasingly stressful work environment.

The results of this study point to the potential impact that wellbeing courses can have on graduate students’ mental health and wellness. Strategies that showed the greatest percent change, such as narrative writing, likely showed such a large change because they were not well-known amongst the students prior to the course’s commencement. Given the dramatic increase in the percentage of students that felt confident applying this strategy from pre- to post-course, these strategies are evidently easy to learn and implement.

The results illuminate the effectiveness of student wellbeing courses and are consistent with a similar study that focused on medical students [27]. Furthermore, this work reflects students’ appreciation and enjoyment of the course. Of the students who responded to the question about course improvements, more than half of participants stated that it should be longer, and one respondent even felt that it should be mandatory. This feedback emphasizes students’ recognition of the importance of the course content and demonstrates their eagerness to learn more about self-care strategies and create cultures of wellbeing.

The 5-month post-course follow up survey highlights the sustainability of the course teachings. With 78.5% of respondents stating that they apply the strategies they learned in this course on at least a weekly basis, the course clearly has lasting benefits. Offering a structured wellbeing course could therefore be a viable method for creating a generation of healthcare professionals that prioritize wellness, recognize the importance of self-care, and have the tools needed to manage their stress.

Assessing wellbeing outcomes at baseline in both 2019 and 2020 allowed for a greater understanding of the impact the COVID-19 pandemic had on student wellbeing. The sharp increase in self-reported burnout from 2019 to 2020 is reflective of the COVID-19 pandemic’s detrimental effect on student mental health and healthcare worker mental health that has been seen around the world [7,8,18].

Many of our results, however, showed better wellbeing amongst the 2020 cohort compared with the 2019 cohort. For example, the 2020 cohort felt more confident in their ability to handle personal problems, which could be indicative of increased resilience amongst students that started the program a few months into the pandemic. In addition, the 2020 cohort reported a significantly higher level of social activity than the 2019 cohort. This unexpected result could be due to the timing of the two courses—the 2019 course took place at the beginning of the school year, when classmates were just beginning to acquaint themselves with one another, while the 2020 course took place halfway through the program.

Although the COVID-19 pandemic undoubtedly had an impact on the discrepancies between the 2019 and 2020 cohort results, other factors could have influenced these results as well. For example, the timing of the course, the in-person versus virtual delivery, and the course structure could have all contributed to differences in baseline wellbeing. This could be considered a limitation of our study, but not regarding reproducibility, as we would expect an in-person course to have an even greater impact on wellbeing outcomes and confidence applying wellbeing strategies.

Reproduction of the “Culture of Wellness” course is feasible and likely to be effective. Devoting as little as 12 h of in-class time (and 50 h total) can increase learners’ skills and improve wellbeing. Educators can serve as wellbeing champions and leaders of change to deliver this curriculum using cases from *A Doctor’s Dozen: 12 Strategies for Personal Health and A Culture of Wellness* and/or related content. Furthermore, several aspects of the instructor’s teaching style—which resulted in extremely positive course evaluations—can be easily employed by many instructors. For example, students noted that they enjoyed the organized nature of the course and the instructor’s honesty about her own wellness journey. One student mentioned that they appreciated how the instructor greeted every student by name as they joined the virtual classroom.

### 4.3. Limitations

Study limitations include sample size, as the small number of participants limited the external validity of our results. Additionally, the paired *t*-test did not include pre-post responses from all participants. To avoid this issue in future studies, surveys should be administered in such a way that allows the administrator to easily connect pre- and post-responses. Additionally, due to the elective nature of this course, it is possible that there was some degree of self-selection bias; students who decided to take the course may have been struggling with their wellbeing more than students who did not take this elective, or the reverse may be true and those students who are at greatest risk for burnout and need it most may not have recognized the need or signed up. Finally, our study could have been stronger had we included a control group of Master of Public Health students who did not participate in the course.

### 4.4. Implications for Research and Clinical Practice

The wellbeing content presented in this course is applicable to all levels of health learners and practitioners; it is the responsibility of educators and healthcare leaders to incorporate wellbeing teachings into training and practice for all. Future steps in education and research should focus on the dissemination and implementation of wellbeing programs for public health graduate students and for practitioners. Larger sample sizes, as well as long-term follow up, would help to determine the strategies that professionals find to be most helpful during practice.

## 5. Conclusions

Culture of Wellness (PH104) is an elective course initiated during the COVID-19 pandemic that increased Master of Public Health students’ confidence in applying wellbeing strategies, significantly decreased burnout rates, and increased students’ wellness immediately post-course and in a 5-month follow up.

## Figures and Tables

**Table 1 ijerph-21-00590-t001:** Course learning objectives and assessments, as indicated in the course syllabus.

Learning Objectives	Assessments
Compare and contrast “health” and “wellness” within the context of personal and population health	Pre-Class Reflections
Discuss the importance of wellbeing and the impact of burnout on individuals and communities	Class Discussion and Participation
Utilize an improvement process to discuss factors that threaten personal and collective wellbeing	Class Discussion and Participation
Apply evidence-based wellness strategies to construct a Personal Health Improvement Plan (PHIP)	Personal Health Improvement Project
Demonstrate leadership and team skills to propose a community-based Culture of Wellness Plan	Culture of Wellness Team Project

**Table 2 ijerph-21-00590-t002:** Five-month 2020 post-course follow up survey for the “Culture of Wellness” (PH104) course.

Question	Scale	% of Responses
*“How important are the wellbeing strategies learned in PH104 to your personal wellness currently?”*	Not at all	0
	Somewhat	28.6
	Moderately	28.6
	Very	28.6
	Extremely	14.3
*“How confident are you in your ability to apply the wellbeing strategies learned in PH104?”*	Not at all	0
	Somewhat	14.3
	Moderately	21.4
	Very	50
	Extremely	14.3
*“Over the last 5 months, how frequently have you applied the strategies learned in PH104?”*	Not at all	0
	Monthly	21.4
	Weekly	57.1
	Daily	14.3
	More than once a day	7.1

**Table 3 ijerph-21-00590-t003:** Percentage change in 2020 pre- and post-course survey responses and significance regarding strategies.

Strategies	% Change	*p*-Value
Narrative writing	54.5	0.004 *
Leading change	42.9	0.0001 *
Cognitive reframing	35.5	0.002 *
Mindfulness	32.4	0.009 *
Appreciative inquiry	32.4	0.003 *
Self-reflection	30.6	0.002 *
Time management	25.8	0.005 *
Prioritizing purpose	20.5	0.01 *
Resilience	15.8	0.002 *
Relationships	13.5	0.05 *
Healthy eating	4.88	0.3
Emotional intelligence	4.88	0.2
Exercise	−2.2	0.4

* indicates statistical significance (*p* < 0.05).

**Table 4 ijerph-21-00590-t004:** Percentage change in 2020 pre- and post-course survey responses and significance regarding wellbeing outcomes.

Wellbeing Outcome	% Change	*p*-Value
**Burnout**		
Over the last month, to what degree have you felt burned out?	−19.4	0.01 *
**Quality of Life**		
Emotional wellbeing	28.3	0.007 *
Physical wellbeing	17.9	0.006 *
Level of social activity	14.8	0.2
Mental wellbeing	14.5	0.04 *
Spiritual wellbeing	12.3	0.1
Quality of life (overall)	5.26	0.2
**Mindfulness**		
Feeling present or in the moment	19.4	0.003 *
**Perceived Stress**		
Difficulties piling up	−25.9	0.005 *
Unable to control important things in your life	−12.1	0.08
Felt that things were going your way	9.09	0.1
Able to handle personal problems	0	0.5

* Indicates statistical significance.

**Table 5 ijerph-21-00590-t005:** Quality of life pre-course survey results ^1^.

Quality of Life Questions	2019 Mean (n = 54)	2020 Mean (n = 20)	% Change	*p*-Value
Your overall quality of life?	8.83	7.74	−12.3	0.12
Your overall mental (intellectual) wellbeing?	6.00	7.26	21	0.07
Your overall physical wellbeing?	7.71	7.53	−2.33	0.06
Your overall emotional wellbeing?	6.00	6.79	13.2	0.08
Your level of social activity?	6.00	6.32	5.3	0.03 *
Your spiritual wellbeing?	6.75	6.63	−1.8	0.09

* indicates statistical significance (*p* < 0.05). ^1^ The scale ranges from 1 (“as bad as it can be”) to 10 (“as good as it can be”).

**Table 6 ijerph-21-00590-t006:** Mindfulness pre-course survey results ^1^.

Mindfulness Scale	2019 (N = 54) Percentage of Responses	2020 (N = 19) Percentage of Responses	% Change	*p*-Value
Not at all	17	0	−100	0.03 *
Minimally	22	37	68.2	0.09
Somewhat	46	42	−8.7	0.38
Mostly	11	21	90.9	0.14
Completely	4	0	−100	0.19

* indicates statistical significance. ^1^ The answers were in response to the survey question, “Over the last month, to what degree have you felt present in the moment?”.

## Data Availability

Data is unavailable due to privacy.

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
