# Peer review of "Evaluation of a Pilot Wellness Elective for Master of Public Health Students during the COVID-19 Pandemic"

_ijerph, 2024, doi:10.3390/ijerph21050590_

Round 1
Reviewer 1 Report
Comments and Suggestions for Authors
Peer review comments:
It’s great that the authors have attempted to evaluate this wellbeing programme. It’s vital that educators and institutions understand what works to improve outcomes for their student populations, and that programmes are evidence-based.
I have commented on some general and specific challenges with the manuscript which I hope will help the authors to further shape this into a useful paper for policymakers and educators. I do think there’s a considerable number of things to tackle to get this paper ready to publish.
General comments
In general, I think you might be clearer and more succinct from the outset, what the course was for and how it was delivered and evaluated. It becomes clearer in the methods section but I was unsure up to that point what related to what, and it’s a lot of mixed-methodology information to absorb. See more detailed pointers below. I’m not sure your references always well support or evidence the points you’re trying to make and sometimes refer to population level conclusions but taken from small-isolated studies. I could not find any tables, they are not linked to the Supplementary files in any way by title and your methods/results/discussion all overlap somewhat? Critically I think your limitations need to be expanded (under a subtitle for clarity). This was a very small study, with only ten linked pre/post experiences, and also comparing baseline data that isn’t really comparable unless you know how the cohort differed in characteristics. Similarly, you need to discuss whether you were actually capturing the phenomena you discuss with the measures you have used. A final important point is that I cannot see any ethical consideration and this is concerning, but perhaps I haven’t been sent it?
Specific comments
Title – the paper suggests this study is overall about sustainability of the wellbeing effects of the intervention, but this is not the focus of the discussion or conclusion. It isn’t even in the main aim of the study (L 106) to examine if effects are sustainable over time? Would it be better simply called Evaluation of a Pilot Wellness Elective for Master of Public Health Students During the COVID-19 Pandemic?
Abstract- You have used a neat mixed methods model to evaluate the impact of the course (asking about its impact on students from a number of different angles), but what the course is and exactly what you’re measuring is not clearly conveyed to the reader.
L 16 I found this sentence confusing – are they students in the class of 2020? Or 2019? Was the elective to learn new wellbeing strategies and these are the methods for evaluating? The whole sentence could be clearer.
L22 78.5% sounds a bit odd for only 22 students? Maybe more than three quarters of respondents (78.5%)
L23 The limitations sentence doesn’t make sense? Differing course structure? Was it different in the pandemic? Are you comparing a course that ran before the pandemic with a course that ran during the pandemic? This does become a bit clearer later in the manuscript but not here. I also think it might be better to highlight the very small sample size as limitations in the abstract.
L24 I think this is overstating what you found with this analysis- it’s a potential indicator that this course might be effective for student confidence, subjective wellbeing etc but you have many limitations and a small sample to contend with. Also, feasibility is something completely different and you would need to have assessed that from a delivery perspective? Caveat this sentence?
Introduction –
L32 For an international audience I think it’s worth defining ‘graduate student’.
L34 I’m not sure reference 2 points to evidence for prevalence of MH problems, it’s about interventions? And it’s almost six years old. I would double check that your references are supporting evidence for you statements and update where you can? 10/20 references are >5 years old. This nature article may be useful – it relates to the largest US study that has been tracking prevalence of student mental health issues (HEALTHY MINDS STUDY) and generated many papers https://www.nature.com/articles/d41586-020-02439-6
L 55 onward - You mention several studies examining student wellbeing interventions but don’t critically evaluate any of them? Are they small studies and samples in isolated colleges, are the measures they used appropriate etc? What are the best methods to evaluate these programmes? Are there any systematic reviews? Would be good to discuss these issues rather than simply detail other small studies?
L102 I’m not sure this is clear. Did it just ‘improve students understanding and knowledge’ rather than ‘advance their capacity to model…’?
L105 Didn’t you also collect follow up data? Was that in your aim? Your title suggests sustainability was key?
L100 Could you make clear whether the US was in lockdown at this point or not i.e. when the course was running? I’m afraid I become a little confused at this point. Did the online course in Covid replace a similar course before Covid? It might be better to explain that here?
Methods
L110 In response to the Covid19 pandemic?
L 119 I have no tables so again this is all unclear? If Table 1 is Supplementary 1 (I think it must be but L138 also says Supplementary 1?) where did the word deliverables come from? This is not mentioned in the manuscript. Similarly course content is based on a book that has no citation?
L136 Assessed by whom and how exactly? Graded? For example how was participation assessed- students had to attend every session? In Supplementary 2 I don’t understand what given a prompt to respond with a paper means? Were the presentations assessed?
L144 Is this a validated Wellbeing measure? I can’t see any reference? If not you need to say why you chose these items and scale?
L152 Do you mean students were tracked using four survey methods? The four surveys offered throughout the course can be divided into 4 sections: (1) wellness strategies, (2) wellbeing outcomes, (3) course evaluations, and (4) a 5-month post-course follow-up.
L158-164 It’s only here I start to understand that you are comparing baseline wellbeing data from another course? Are these the same students – I think this needs to be explicit? Is one course face to face in 2019, and the other course delivered online during the pandemic in 2020– but a version of the face to face course? Why wasn’t the Culture of Wellbeing Course mandated like the original? Did you collect any student characteristics to be able to match students? Otherwise, how do you control for different students taking each course?
None of the measures you’ve used have been attributed/cited i.e. short quality of life scale
L189 No idea where table 2 is? Can’t even see anything in Supplementary that might correspond?
L93 You discuss the evaluation link being anonymous but also being able to match 10/22 students? How, if they were anonymous? Did you mention this in ethics (I can’t see your ethics approval I’m afraid).
L195 Better to say ‘we carried out t-test analyses with…’
Results – Would be better just to describe each set of results in turn rather than do an introduction to results.
I also think 'how many students took part in each survey' would be better presented as a table? Do we know anything about the differences between all these students, gender, age?
L205 Results are not the place to describe why you do something, that’s methods. Results are just results. You need to report any t-test findings correctly using brackets etc.
Tables 3 and 4 not here? And also don’t appear to be in Supplementary? See comments above.
L242 onward- I would avoid saying ‘56% said…’ or similar when you have so few responses. Maybe ‘more than half (n=10?)’
Would suggest there are too many comments cited here. Just use a couple of examples and keep it tighter. How did you choose what to highlight?
L279 onward- again you are describing methods in results. This whole section is methods with the exception of line 290.
Without any tables I’m afraid I can’t comment on the rest of the results.
L341 How do you identify what is a ‘key’ quote? Without explaining in the methods how you have systematically gone through your open text data to assess it– you could surely be simply cherry picking?
Discussion – First paragraph is introduction material again? I strongly recommend that you don’t talk about ‘greatest percent change’ – this was a very small sample. The biggest differences were seen maybe? I would suggest you caveat your language…this study points to, there are indicators here, this might suggest..
L380-389 As before you seem to be reporting findings in the discussion?
I’m a little concerned about the focus on the instructor’s teaching style? It’s very personal and positive. Is there a conflict of interest here?
I don’t think it’s good enough to say ‘our sample was small, and to avoid in future…’, I think the language needs to take more responsibility. 'We acknowledge this was a small sample and pragmatic attempt to evaluate a wellbeing course'.
Conclusion I would write a full conclusion or don’t include one.
I realise this is a very long list of suggestions and it's not exhaustive. I have really carefully considered your manuscript as best I can with all the information I had. I do hope my comments are helpful. They are meant to be!
Comments on the Quality of English LanguageSome robust editing/proofing is needed.
Author Response
Dear Reviewer,
Thank you for your thoughtful and timely feedback. My colleagues and I have carefully revised our manuscript in response to your comments as well as feedback from two other reviewers. We have attached the revised manuscript, including updated Tables and Figures, for your review.
Below are some highlights of the changes we have made:
General Comments:
- Grammar and Clarity: We made extensive grammar corrections throughout the abstract and the rest of the manuscript.
- References: We have updated our references section to include recent publications relevant to our research.
- Integration of Tables and Figures: We have incorporated our tables and figures directly into the manuscript to improve readability and coherence.
- Limitations Section: We have significantly expanded our discussion of the study's limitations.
- Ethical Considerations: We added a section on "Ethical Approval and Consent," detailing our IRB approval, exemptions, and our process for obtaining informed consent.
Specific Comments:
- Manuscript Title: We revised our title following your insightful suggestion, which better reflects the content and focus of our study.
- Clarification of Data Cohorts: In our original manuscript, it was unclear whether we referred to the class of 2020 or 2021. We have now revised the abstract and relevant sections to clarify this distinction.
- Abstract Improvements: Following your recommendations, we have restructured our abstract for better clarity and expanded it to provide a more comprehensive overview of our study.
- Introduction: Significant revisions include the incorporation of the nature reference you suggested, thorough proofreading, and an expanded description of our "Culture of Wellness (PH104)" course.
- Methods Section: We have expanded this section to address your suggestions (e.g. described class of 2020 vs class of 2021; clarified the point of “The SBDoH course”), cited our validated self-reported measurements more clearly, and included additional tables as requested. Furthermore, we have clarified our methodology for analyzing participant comments.
- Results Section: We removed the initial presentation of results and added a new subsection on "Participants," which includes detailed demographics. Also, we revised our paragraph subtitles.
- Discussion Section: We reformatted this section into four distinct parts—summary of results, results in context, limitations, and implications—based on your advice. Additionally, we have relocated certain descriptive phrases to the results section to maintain focus and flow.
- Conclusion Section: We revised and expanded this section based on your suggestions.
On behalf of all co-authors, I express our sincere gratitude for your detailed and constructive feedback. We are confident that these revisions have significantly strengthened our manuscript.
Thank you once again for your time and support.

Reviewer 2 Report
Comments and Suggestions for Authors
Dear Authors, thank you for your research that definitely addresses important issues of mental functioning of students and their well-being. You rpopose somewhat interesting approach to modification of students well-being. Still, there are some comments on your paper:
1. It was really complicated to go htrough the design. It would be better, if you would make it as some scheme or table - first they did the survey, in some weeks they did next task and so on.
2. I couldn't find clear description of the sample. You mention 22 students - what age? what gender? what year of education? what major?
3. Than, you mention some assessment in 2019 - did ou mean assessment of the same students, or different students? If those were different students, how many? age? year?
4. There is generally no conclusion, it is too short and too generalized.
Author Response
Dear Reviewer,
Thank you for your thoughtful and timely feedback. My colleagues and I have carefully revised our manuscript in response to your comments as well as feedback from two other reviewers. We have attached the revised manuscript, including updated Tables and Figures, for your review.
Below are some highlights of the changes we have made:
General Comments:
- Grammar and Clarity: We made extensive grammar corrections throughout the abstract and the rest of the manuscript.
- References: We have updated our references section to include recent publications relevant to our research.
- Integration of Tables and Figures: We have incorporated our tables and figures directly into the manuscript to improve readability and coherence.
- Limitations Section: We have significantly expanded our discussion of the study's limitations.
- Ethical Considerations: We added a section on "Ethical Approval and Consent," detailing our IRB approval, exemptions, and our process for obtaining informed consent.
Specific Comments:
- We created several tables (referenced and attached to the document) that would explain our study design, methods, and results.
- We received limited data on the participant demographics. However, we added a "Participant" section under results that describe participant demographics.
- Course that was offered in 2019 consisted of a sample of students from a different cohort; compared to the sample of students in the course offered in 2020. We clarified this under methods.
- We expanded our conclusion section based on your feedback
Additional revisions:
- Manuscript Title: We revised our title following your insightful suggestion, which better reflects the content and focus of our study.
- Clarification of Data Cohorts: In our original manuscript, it was unclear whether we referred to the class of 2020 or 2021. We have now revised the abstract and relevant sections to clarify this distinction.
- Abstract Improvements: Following your recommendations, we have restructured our abstract for better clarity and expanded it to provide a more comprehensive overview of our study.
- Introduction: Significant revisions include the incorporation of the nature reference you suggested, thorough proofreading, and an expanded description of our "Culture of Wellness (PH104)" course.
- Methods Section: We have expanded this section to address your suggestions (e.g. described class of 2020 vs class of 2021; clarified the point of “The SBDoH course”), cited our validated self-reported measurements more clearly, and included additional tables as requested. Furthermore, we have clarified our methodology for analyzing participant comments.
- Results Section: We removed the initial presentation of results and added a new subsection on "Participants," which includes detailed demographics. Also, we revised our paragraph subtitles.
- Discussion Section: We reformatted this section into four distinct parts—summary of results, results in context, limitations, and implications—based on your advice. Additionally, we have relocated certain descriptive phrases to the results section to maintain focus and flow.
- Conclusion Section: We revised and expanded this section based on your suggestions.
On behalf of all co-authors, I express our sincere gratitude for your detailed and constructive feedback. We are confident that these revisions have significantly strengthened our manuscript.
Thank you once again for your time and support.

Reviewer 3 Report
Comments and Suggestions for Authors
Dear Authors,
Thank you for the opportunity to review your manuscript. It addresses the important issue of well-being, which has been significantly affected during the Covid-19 pandemic.
I found it intriguing that in the USA, graduates are more susceptible to psychological discomfort, but I did not learn why. It would be helpful to indicate this in the Introduction.
The study you conducted could be termed a quasi-experiment. It would be beneficial to include this in the study description.
I wonder if a 12-hour course is sufficient to make long-term changes in the behavior of participants. Therefore, it might be better to emphasize that this is a pilot study and consider extending the course in subsequent years.
I do not understand the purpose of this: "The SBDoH course was a required 10-week course that featured three main themes, personal wellbeing, public health, and population health. The SBDoH course was completed by 54 students, compared to 22 students in the 'Culture of Wellbeing' course." What was the aim of this comparison?
How do we know that the questions were answered by students participating in the course?
Content in points 2.7 and 3 seems to repeat itself. In point 2.7, please specify which statistical tests were used and what significance level was adopted. This is the place for such information.
Is there a numbering error? - 3.12020. Pre-Post Course Strategies and Wellbeing Surveys
Is it appropriate to present percentage values with such small sample sizes?
The conclusions drawn from the research are very brief. It would be beneficial to expand upon them.
The idea for the course is excellent but requires further clarification and extension, as pointed out by the respondents. It is important for as many students as possible to benefit from it.
Author Response
Dear Reviewer,
Thank you for your thoughtful and timely feedback. My colleagues and I have carefully revised our manuscript in response to your comments as well as feedback from two other reviewers. We have attached the revised manuscript, including updated Tables and Figures, for your review.
Below are some highlights of the changes we have made:
General Comments:
- Grammar and Clarity: We made extensive grammar corrections throughout the abstract and the rest of the manuscript.
- References: We have updated our references section to include recent publications relevant to our research.
- Integration of Tables and Figures: We have incorporated our tables and figures directly into the manuscript to improve readability and coherence.
- Limitations Section: We have significantly expanded our discussion of the study's limitations.
- Ethical Considerations: We added a section on "Ethical Approval and Consent," detailing our IRB approval, exemptions, and our process for obtaining informed consent.
Specific Comments:
- Given that our study was conducted during COVID-19, we wanted to compare the data during COVID-19 to pre-COVID. Course SBDoH offered in 2019 has the same pre-survey questions compared to our Culture of Wellness (PH104) course offered in 2020. Therefore, we decided to use this data compare responses to questions related to burnout, mindfulness, etc.. We clarified and expanded this section under Methods section 2.4
- We know that questions were answered by students participating in the course because we provided the survey to only students enrolled in the course. However, survey was anonymous and they received informed consent.
- We removed repeated content in sections 2.7 and 3. Also, explained in the Methods section 2.7 that "While for such a small sample size, a larger significance level might be preferred, based on the actual results from 3.2, we decided to set the significance level to 0.05 or 5% as the “boundary” between “significant” vs “not significant.”" We received consultation from Statistician.
- We renamed this section (was 3.1 2020 ...) to section 3.2 "Pre-Post Course Strategies and Wellbeing Surveys"
- We revised the way we presented our data to avoid presenting percentage values with such small sample sizes. Or in some cases, we provided sample number (n = ) next to percentages.
- We revised and expanded our conclusion section.
Additional revisions:
- Manuscript Title: We revised our title following your insightful suggestion, which better reflects the content and focus of our study.
- Clarification of Data Cohorts: In our original manuscript, it was unclear whether we referred to the class of 2020 or 2021. We have now revised the abstract and relevant sections to clarify this distinction.
- Abstract Improvements: Following your recommendations, we have restructured our abstract for better clarity and expanded it to provide a more comprehensive overview of our study.
- Introduction: Significant revisions include the incorporation of the nature reference you suggested, thorough proofreading, and an expanded description of our "Culture of Wellness (PH104)" course.
- Methods Section: We have expanded this section to address your suggestions (e.g. described class of 2020 vs class of 2021; clarified the point of “The SBDoH course”), cited our validated self-reported measurements more clearly, and included additional tables as requested. Furthermore, we have clarified our methodology for analyzing participant comments.
- Results Section: We removed the initial presentation of results and added a new subsection on "Participants," which includes detailed demographics. Also, we revised our paragraph subtitles.
- Discussion Section: We reformatted this section into four distinct parts—summary of results, results in context, limitations, and implications—based on your advice. Additionally, we have relocated certain descriptive phrases to the results section to maintain focus and flow.
- Conclusion Section: We revised and expanded this section based on your suggestions.
On behalf of all co-authors, I express our sincere gratitude for your detailed and constructive feedback. We are confident that these revisions have significantly strengthened our manuscript.
Thank you once again for your time and support.

Round 2
Reviewer 3 Report
Comments and Suggestions for Authors
Dear Authors, thank you for improving the manuscript. Now your examination has a clear appearance, is legible and understandable. Congratulations on your idea and research. I hope you won't stop piloting.